# Integrative multi-region molecular profiling of primary prostate cancer in men with synchronous lymph node metastasis

Udit Singhal [1,2,3,4,13] ✉, Srinivas Nallandhighal [1,13], Jeffrey J. Tosoian [5,6], Kevin Hu[7], Trinh M. Pham[1], Judith Stangl-Kremser[1,8], Chia-Jen Liu[9], Razeen Karim[9], Komal R. Plouffe[4,7], Todd M. Morgan [1,3], Marcin Cieslik[3,4,7], Roberta Luciano[10], Shahrokh F. Shariat[8], Nadia Finocchio[11], Lucia Dambrosio[11], Claudio Doglioni [10], Arul M. Chinnaiyan [1,3,4,7,12], Scott A. Tomlins[7], Alberto Briganti[11,14], Ganesh S. Palapattu [1,3,8,14], Aaron M. Udager [3,4,7,14] ✉ & Simpa S. Salami [1,3,4,14] ✉

Localized prostate cancer is frequently composed of multiple spatially distinct tumors with significant inter- and intra-tumoral molecular heterogeneity. This genomic diversity gives rise to many competing clones that may drive the biological trajectory of the disease. Previous large-scale sequencing efforts have focused on the evolutionary process in metastatic prostate cancer, revealing a potential clonal progression to castration resistance. However, the clonal origin of synchronous lymph node (LN) metastases in primary disease is still unknown. Here, we perform multi-region, targeted next generation sequencing and construct phylogenetic trees in men with prostate cancer with synchronous LN metastasis to better define the pathologic and molecular features of primary disease most likely to spread to the LNs. Collectively, we demonstrate that a combination of histopathologic and molecular factors, including tumor grade, presence of extra-prostatic extension, cellular morphology, and oncogenic genomic alterations are associated with synchronous LN metastasis.

Prostate cancer, the most commonly diagnosed solid organ malignancy and the second leading cause of cancer-related death among men in the developed world, is a heterogenous disease[1]. The clinical trajectory and response to therapy at all stages is variable, and the known molecular drivers of prostate cancer are generally myriad[2–4]. In primary, localized prostate cancer, where multifocality is common, genomic heterogeneity is frequent[4–7]. However, while both intra- and inter-tumoral genomic heterogeneity have been linked to aggressive biologic behavior, the identification of the biologically dominant clone that drives aggressive disease has remained elusive[5,6,8,9]. For a given

[1]Department of Urology, Michigan Medicine, Ann Arbor, MI, USA. [2]Department of Urology, Mayo Clinic, Rochester, MN, USA. [3]Rogel Cancer Center, Michigan Medicine, Ann Arbor, MI, USA. [4]Michigan Center for Translational Pathology, Michigan Medicine, Ann Arbor, MI, USA. [5]Department of Urology, Vanderbilt University, Nashville, TN, USA. [6]Vanderbilt-Ingram Cancer Center, Nashville, TN, USA. [7]Department of Pathology, Michigan Medicine, Ann Arbor, MI, USA. [8]Department of Urology, Medical University of Vienna, Vienna, Austria. [9]College of Literature, Science, and Arts, University of Michigan, Ann Arbor, MI, USA. [10]Department of Pathology, Universita Vita-Salute San Raffaele, Milan, Italy. [11]Department of Urology, Universita Vita-Salute San Raffaele, Milan, Italy. [12]Howard Hughes Medical Institute, University of Michigan, Ann Arbor, MI, USA. [13]These authors contributed equally: Udit Singhal, Srinivas Nallandhighal. [14]These authors jointly supervised this work: Alberto Briganti, Ganesh S. Palapattu, Aaron M. Udager, Simpa S. Salami. ✉e-mail: usinghal@med.umich.edu; udager@med.umich.edu; simpa@med.umich.edu

patient with primary multifocal prostate cancer, knowledge of which clone (or clones) is most likely to give rise to metastasis has the potential to inform management, including for interpretation of tissue-based prognostic tests, selection of precision treatment approaches, and development of novel therapeutic strategies.

Historically, the "index tumor" has been defined as the tumor with the largest volume and/or highest grade and presumed to drive prognosis[10]. However, in a proportion of cases, the largest tumor focus is discordant with the highest grade tumor or the tumor associated with adverse pathologic characteristics, such as extraprostatic extension (EPE) or seminal vesicle invasion (SVI)[11]. Additionally, in prostate specimens where the largest focus shares the same histologic grade (i.e., Grade Group (GG)) as other foci, it is unknown if the largest focus is molecularly unique and/or possesses the highest probability of spread. Further, a relatively low-grade region of an overall high-grade focus can periodically give rise to lethal clones[12]. Thus, the concept of the biologically dominant clone in primary prostate cancer remains poorly defined. High-throughput genomic profiling with next-generation sequencing (NGS), however, has facilitated a better understanding of the molecular drivers of cancer dissemination as well as identified distinct tumor clones associated with metastasis in breast, colorectal, lung, liver, kidney, and hematologic cancers[13–18]. In metastatic, castrate-resistant prostate cancer, the lethal clone largely arises from a single monoclonal precursor cell population, with continued seeding from a single, dominant lineage post-metastasis[19–21]. In primary, multifocal disease, large-scale transcriptomic and genomic profiling efforts have revealed distinct expression profiles among separate foci and the presence of multiple genetically discrete cancer cell clones suggesting independent origins[6–8,22]. However, to date, comprehensive analyses of primary tumors inclusive of multifocal disease with matched metastatic samples to delineate the lesions most likely to metastasize have been limited.

Here, we perform multi-region molecular profiling of primary prostate cancer and synchronous lymph node (LN) metastasis to better understand what defines the biologically dominant clone in this setting. By analyzing both primary tumor foci and matched LN metastases, we attempt to build upon the existing foundation of knowledge regarding the heterogeneity of multifocal prostate cancer. An improved comprehension of the molecular underpinnings that drive LN metastasis in this setting would complement the existing breadth of data elucidating clonal evolution in metastatic, castrate-resistant disease and allow for improved risk stratification and treatment allocation in the localized setting.

## Results

### Mapping primary prostate cancer to synchronous LN metastasis

From a cohort of 18 patients (age 50–73 years) who had radical prostatectomy and pelvic lymph node (LN) dissection for prostate cancer, we performed multi-region sampling from formalin fixed paraffin embedded (FFPE) blocks yielding 103 primary tumor and 28 LN metastasis samples (patient data is summarized in supplementary file – **Source Data**). We performed high depth, targeted, multiplex DNA sequencing to characterize the genomic profile of each tumor region using two targeted DNA NGS panels: the Comprehensive Cancer Panel (CCP; 409 genes and 15,992 amplicons) and a custom Pan-GenitoUrinary (Pan-GU) cancer panel (135 genes and 3127 amplicons). Targeted RNA NGS sequencing was also performed using a custom prostate cancer-focused NGS panel to evaluate the gene fusion status of each sample and derive relevant tissue-based prognostic scores [Myriad Prolaris™ Cell Cycle Progression (mxCCP) score, Oncotype DX™ Genomic Prostate Score (mxGPS), and Decipher™ Genomic Classifier (mxGC)] as previously performed[7] (Fig. S1). We determined and compared histopathological characteristics, RNA tissue-based prognostic signatures, somatic DNA mutations, copy number alterations (CNA), and gene fusion status between primary and LN disease (Fig. 1).

After quality control (QC) steps as described in the methods, 10 patients (65 primary tumor and 16 LN metastatic samples) had sufficient data for phylogenetic analyses. All patients in the analytic cohort exhibited adverse pathological characteristics, including cribriform, single cell, or solid cell pattern in at least one tumor focus. A total of eight patients had EPE, one demonstrated SVI and four showed evidence of lymphovascular invasion (LVI). We noted substantial transcriptomic heterogeneity, including discordant-derived prognostic gene signature scores both within lesions and between lesions in the same patient (Fig. 1). Recurrent CNAs were assessed from targeted NGS data using an approach that was previously benchmarked to fluorescence in situ hybridization (FISH), comparative genomic hybridization (CGH) array, and whole exome sequencing (WES) data, showing a high degree of concordance among these approaches[23]. In a subset of samples in our cohort (n = 8 samples; from patient #1), we also performed low-pass whole genome sequencing (LPWGS) and found high concordance with recurrent CNAs identified using targeted DNA NGS such as 8p loss and 8q gain (Fig. S2). TMPRSS2:ERG fusion transcripts were identified in four patients and the SLC45A3:ERG fusion transcripts were detected in one patient (Fig. 1, Fig. S1). Phylogenetic analyses were performed with R phangorn package using neighbor joining method to computationally reconstruct the clonal evolution of prostate cancer and determine the likely clonal origin of LN metastasis for each patient (Fig. 2, S3-10)[19,24]. Additionally, we utilized PhyloWGS as an orthogonal method to generate phylogenetic reconstructions in a subset of patients (n = 3) and observed generally concordant results (Fig. S3 and Fig. 2). Our data reveal several possible histopathologic and molecular factors associated with disease spread as described below.

### Metastatic spread of prostate cancer is linked to aggressive pathologic features

Several pathologic features were associated with LN metastasis, including histologic grade, cribriform or single cell pattern, and EPE (Fig. 1). For example, all 10 patients had at least one primary tumor region with a cribriform pattern, and cribriform pattern was observed in both the dominant primary tumor region and LN metastasis in seven patients. Interestingly, a single cell pattern – associated with high histologic grade – was detected in LN specimens in only two patients (20%), though it was present in the primary tumors of eight patients. Intriguingly, in 8 patients with EPE, phylogenetic reconstruction supported the region of EPE as the likely source of the LN metastasis in four cases (Fig. 2a, S4,7,8). In patient #1 (Fig. 2a and S3a), all tumors harbored a TMPRSS2-ERG gene fusion. While four primary tumor regions showed concordant TP53, IL6ST, and TPR mutations, only two primary tumor regions (P1 and P2, both EPE) also harbored an LRP1B mutation and high-level CNAs (e.g., like 16q loss and 8p12 loss) that were also present in the two LN (LN1 and LN2) metastasis foci. Notably, primary tumor regions P3 and P4, with the presence of single cells, did not appear to have seeded the LN metastatic foci. In this patient, primary tumor regions P1 and P2 (GG5 regions with EPE) were most likely the source of LN metastasis. In patient #2 (Fig. S4), all tumor regions were ETS gene fusion negative. A CDK12 frameshift mutation with MYC and FGFR3 gain was also detected in all regions. Phylogenetic analysis suggests P4 (a focus with EPE) most closely resembles the LN1 metastasis. In patient #34 (Fig. S7), BRCA1 and PTEN losses were seen in P1, P4, P7, P8, P9, LN1, LN2, and LN3, but not in P2, P3, and P5, suggesting two different branches of clonal evolution. Regions P4 and P8 showed evidence of EPE. Phylogenetic analysis suggests that regions P1, P4, P7, P8, and P9 are most likely the source of LN metastasis in this patient. In patient #38 (Fig. S8), regions P3, P4, P5, P6, and P7 (a region of EPE) displayed FOXA1 mutations along with losses of PTEN and RB1, similar to the LN foci (LN1). The area of P6 and LN1 shared additional

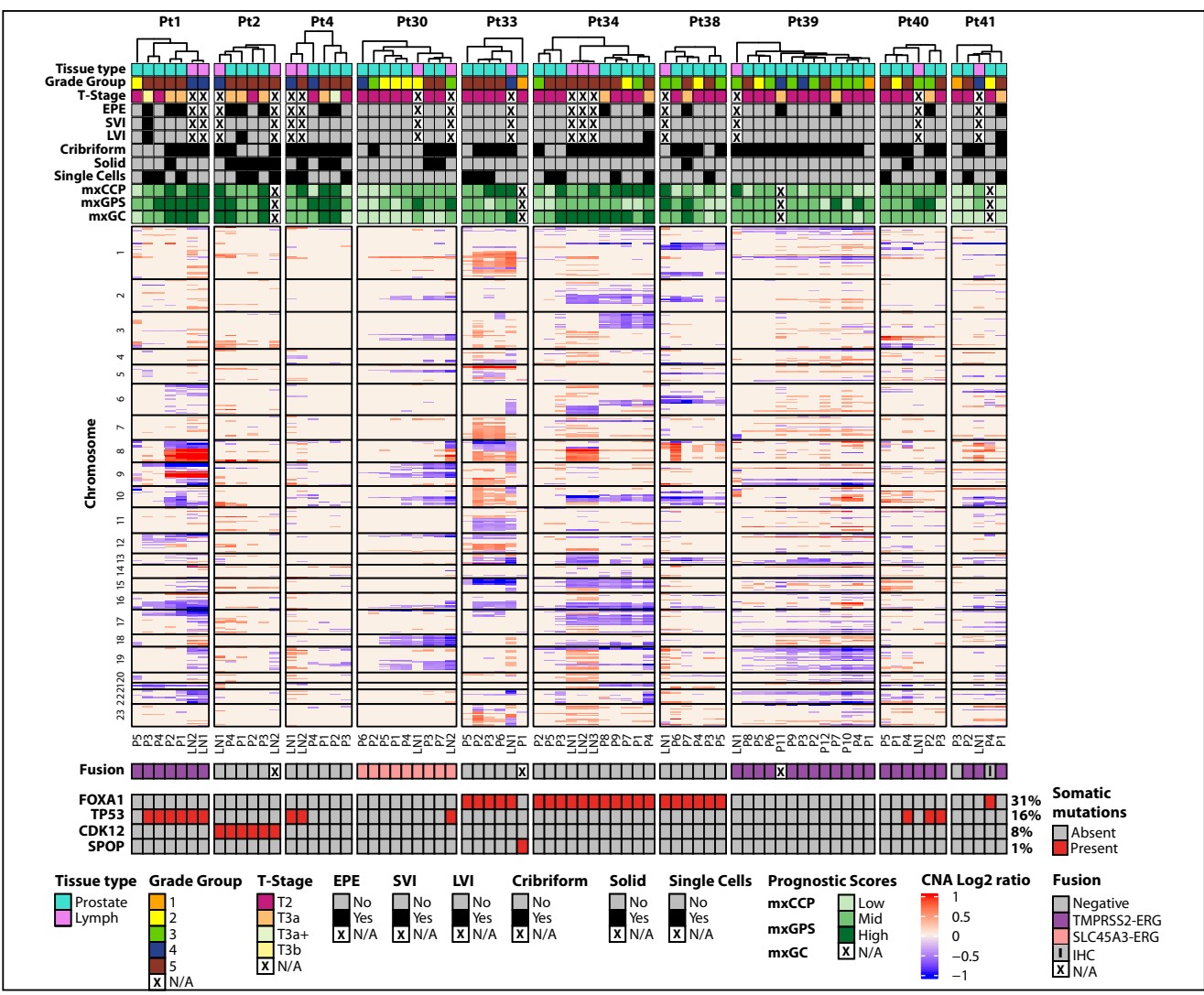

**Fig. 1 | Integrative comprehensive multi region genomic profiling of primary prostate cancer with synchronous lymph node (LN) metastasis.** We used two targeted DNAseq panels to identify key somatic mutations and copy number alterations (CNA) across 10 patients who passed our custom quality control filtering criteria. For CNA analysis, only the top 445 genes (no. of amplicons per gene > 4 & $\log_{10}$ false discovery rate <0.01 & absolute $\log_2$CNvalue > 0.3) with losses and gains are displayed in the heatmap. Unsupervised hierarchical clustering of all tumor regions within each patient was performed to interrogate primary tumor regions that cluster with their respective synchronous LN metastasis regions using $\log_2$ normalized data. Genes were ordered by the chromosome number along with their start and end positions within each chromosome. CNA for the known prostate cancer-relevant genes are annotated. *ETS* gene fusion status was derived from targeted RNAseq data using an in-house fusion quantification pipeline. Relevant clinicopathologic variables such as grade, stage, extraprostatic extension (EPE), seminal vesicle invasion (SVI), lymphovascular invasion (LVI), cribriform pattern, solid pattern, single cells and derived commercially available prognostic scores (mxCCP (derived Cell Cycle Progression score or Prolaris™), mxGPS (derived Genomic Prostate Score or Oncotype™), and mxGC (derived Genomic Classifier or Decipher™) for each sample are annotated on the heatmap. Prognostic scores were categorized into low, mid, and high groups based on their Q1 and Q3 values for comparison among samples with the same patient as well as comparison across different scores within each sample. We observed intra- and inter-patient heterogeneity in histologic grade, genomic alterations, and derived prognostic gene signatures. Source data are provided as a Source Data file.

loss of *CDKN2B* and gain of *RECQL4*, suggesting this as the most likely clonal source of the LN metastatic focus, though the region with EPE (P7) likely contributed given its shared mutational burden and proximity to P6 and LN1 by phylogenetic analysis. Taken together, these findings suggest that aggressive pathologic features – EPE, histologic grade, and/or cribriform pattern – may be associated with the development of LN metastases.

### Role of driver alterations and genomic complexity in prostate cancer metastasis to LNs

As shown in Fig. 1, established early oncogenic driver alterations including *ERG* gene fusions as well as *FOXA1*, *CDK12*, and *SPOP* mutations were typically present in both the primary tumor and LN metastatic foci. The presence of multiple such alterations within three

patients in our cohort (Fig. 2 and S7) suggests the possibility of tumor multiclonality – a well-established concept in prostate cancer[5,6,9]. Importantly, in these patients, NGS analyses only showed evidence of a single early oncogenic driver in the LN metastases, supporting the monoclonal theory of metastatic prostate cancer (see below). Additionally, our analyses revealed substantial intratumoral heterogeneity across the sampled primary tumor regions, including *TP53* mutations and genomic complexity. Intriguingly, although *TP53* mutations were identified in multiple primary tumor regions from two patients (#1 and #40, Fig. 2 and S10), this mutation was only observed in the LN metastatic foci from patient #1. Conversely, *TP53* mutations were detected only in the LN metastatic foci from two other patients (#4 and #30, Fig. S5 and S6). Regardless, these data suggest that *TP53* mutations may be associated with metastatic progression of primary prostate cancer.

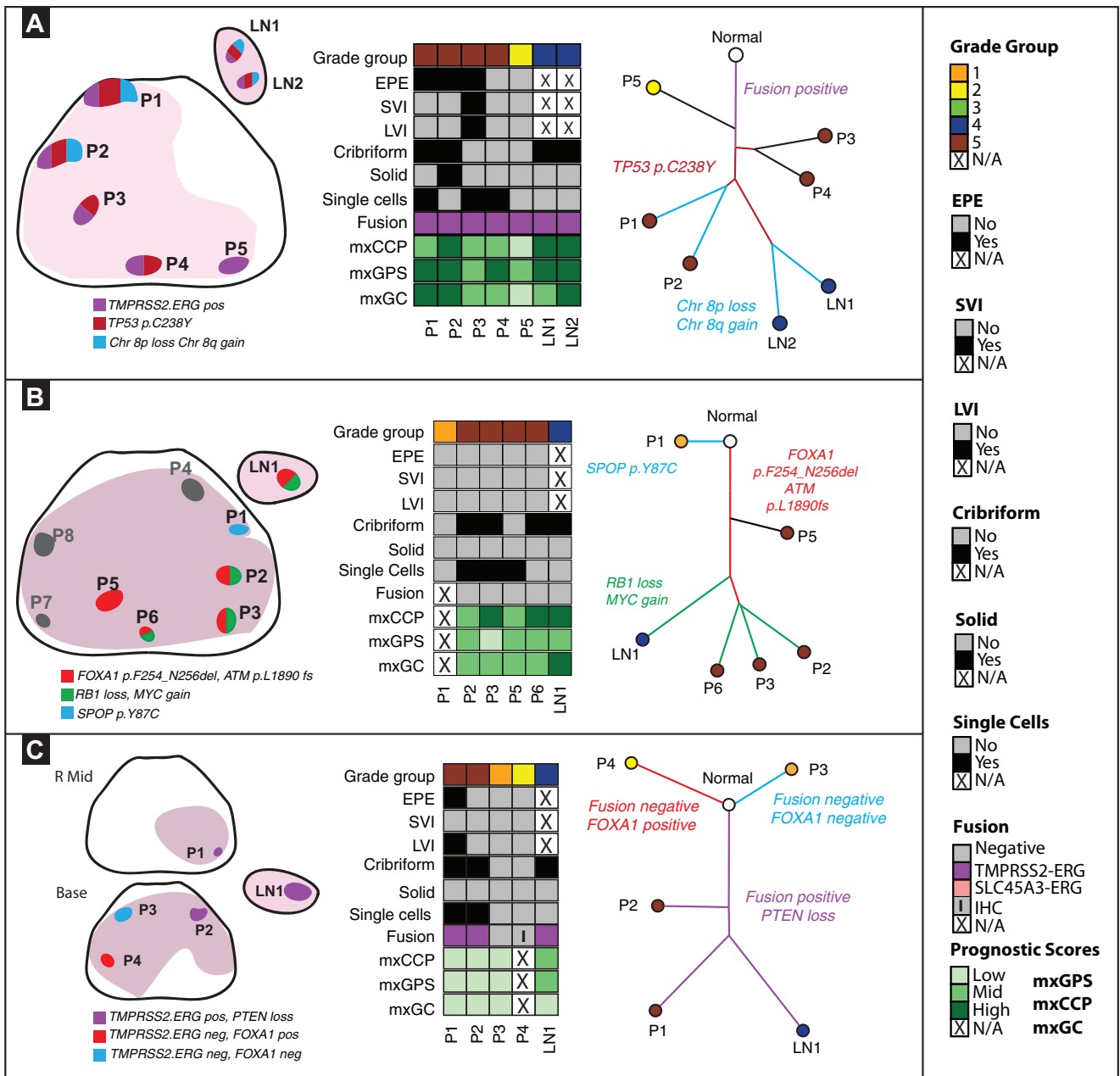

**Fig. 2 | Phylogenetic evolutionary analysis to determine the biologically dominant nodule.** Source data are provided as a Source Data file. **A. Spatial spread linked to extra-prostatic extension (EPE). Left panel:** In patient #1, 5 primary and 2 LN regions encompassing Grade Groups (GG) 2-5 were analyzed. We identified three distinct molecular subtypes. **Middle panel:** All tumor regions, including the two LN metastases regions, were *TMPRSS:ERG* fusion-positive and with heterogeneous prognostic gene signatures. **Right panel:** *TP53* somatic mutation was detected on all tumor regions, except for P5. Key chromosomal alterations, such as 8p loss and 8q gain, were only detected in regions P1, P2, LN1, LN2. Here, the data suggest that primary tumor regions P1, P2 closely resemble LN1, LN2. Additionally, P5 represents a distinct subclone unrelated to other primary regions or LNs. **B. Driver alterations in metastasis to LN. Left panel:** In patient #33, 8 primary tumor (GG1-5) and 1 LN metastasis region were analyzed. Gray regions (P4,7,8) designate those with low tumor purity that were excluded. We identified three molecular subtypes. **Middle panel:** All regions were *ETS* fusion negative except for P1, P8. P4, P7, and P8 regions were excluded due to low tumor content. **Right panel:** An *SPOP* mutation was detected only in P1. Driver alterations: *FOXA1, ATM* frameshift mutations were detected in regions P2, P3, P5, P6, LN1. *RB1* loss and *MYC* gain were seen in P2, P3, P6, and LN1, but not P5. Here, data suggest that primary regions P2, P3, P6, all GG5 regions with cribriform patterns, closely resemble LN1. **C. LN metastasis in multiclonal primary disease. Left panel:** In patient #41, 4 primary (GG2-5) and 1 LN metastasis region were analyzed. P1, a region of EPE, was taken from the mid prostate and P2-P4 was taken prostate base. We identified three distinct tumor clones. **Middle panel:** All samples except P3 and P4 were *TMPRSS:ERG* fusion-positive. **Right panel:** P4 had a *FOXA1* somatic mutation which was not seen in other regions. Here, data suggest that primary region P1 closely resembles LN1, as both are *TMPRSS:ERG* fusion-positive and harbor *PTEN* loss.

Increased genomic complexity was also frequently observed in LN metastatic foci. For example, in a patient with organ-confined (pT2) prostate cancer and LN metastasis (patient #33, Fig. 2b and S3b), the samples demonstrated genomic complexity with *FOXA1* and *ATM* frameshift mutations detected in some primary tumor and LN metastasis regions (P2, P3, P5, P6, LN1); an *SPOP* mutation was detected only in a GG1 primary tumor focus (P1); and *RB1* loss and *MYC* gain were shared between the presumed dominant primary tumor regions and the LN metastasis focus (P2, P3, P6, LN1). In patient, #41 (Fig. 2c), regions P1, P2 and LN1 all were *TMPRSS:ERG* gene fusion-positive and harbored

*PTEN* loss. P1 most closely resembled LN1 based on phylogenetic analysis, suggesting this as the region that likely gave rise to the LN1 metastatic focus. In the PhyloWGS approach, however, (Fig. S3c), P2 was identified as closely resembling the region of the LN metastasis (LN1). Notably, in this approach, P1 was filtered out by the algorithm despite having the *TLX1* mutation because a clonal loss of 8q was not found in this sample. However, it is noteworthy that regions P1 and P2 appear to be spatially related anatomically (Fig. 2c, *left panel*). In patient #2 (Fig. S4), LN1 and P4 shared a *CDKN2B* loss, along with a *CDK12* frameshift mutation and *MYC* and *FGFR3* loss as discussed above. In patient #30 (Fig. S6), we observed shared loss of *CDKN2A*, *ERCC2*, and *ERCC3* in lesions P1, P3, P4, P7 and LN2. LN2 had an additional mutation in *TP53*. Of all the tumor regions, P3 most resembled LN2 in mutational profile and genomic complexity, with phylogenetic analysis suggesting this as the likely clonal origin of LN2 metastasis. In patient #34 (Fig. S7), *PTEN* and *BRCA1* losses were observed among P1, P4, P7, P8, P9, and LN1, LN2, and LN3. In patient #38 (Fig. S8), P6 and LN1 both displayed *CDKN2B* loss and *RECQL4* gain as discussed above. Finally, patient #4 (Fig. S5) showed significant genomic complexity and additional targeted NGS with a 500-gene panel revealed an *MUTYH* mutation with tumor mutational burden and microsatellite instability in regions P1 and P2, suggesting these regions developed continued genomic aberrations after LN1 and LN2 metastasis (as suggested by earlier clonal branching of the LN metastasis on phylogenetic tree).

Taken together, these data support the possible role of driver genomic alterations in the development of synchronous LN metastasis in primary prostate cancer.

### Lymph node metastatic homogeneity

We observed histologic, genomic, and transcriptomic heterogeneity across the primary prostate cancer foci (Fig. 1, S1). A total of five patients had multiple LN areas analyzed. These synchronous LN metastases were typically homogenous within a given patient, consistent with existing literature regarding metastatic foci[19,20]. Notably, several histopathologic features were associated with LN metastasis, including GG and cribriform or single cell pattern (Fig. 1). In patients with multiple LN foci analyzed, we noted mostly homogenous patterns of mutations, suggesting a shared clonal origin. For example, in patient #1 (Fig. 2a and S3), both LN1 and LN2 shared chr8p loss and chr8q gain, as well as mutations in *TP53*. In patient #4 (Fig. S5), LN1 and LN2 shared loss of *CDKN2B*, *FANCA* gain, and *TP53* frameshift mutations. All three LN metastatic regions LN1, LN2, and LN3 in patient #34 (Fig. S7) shared *PTEN*, *BRCA1*, *BRCA2*, and *CDKN2A* losses as well as *MYC* gain.

In contrast, two of the five patients with multiple LN foci analyzed displayed discordant molecular profiles and thus separate branching in the phylogenetic analysis, including patient #2 (Fig. S4) with LN1 and LN2 showing discordant losses of *CDKN2B* (LN1) and *TP53* (LN2). In patient #30 (Fig. S6), LN2 shared features of *CDKN2A*, *ERCC2*, *ERCC3* loss, and *TP53* mutations, whereas LN1 did not show these shared alterations. While low genomic complexity, early clonal branching or evolution, or low tumor content are possible explanations for these observations, the possibility of LN metastasis from separate tumor regions or clones cannot be excluded.

## Discussion

Here, we performed multi-region, multi-platform molecular characterization of primary prostate cancer to dissect the clonal origin of synchronous LN metastases. First, we observed intra- and inter-tumoral histologic and genomic heterogeneity of primary disease consistent with existing literature[4–7,14]. Notably, however, we found relative homogeneity between LN specimens within a given patient. Second, we demonstrated that while LN metastases are highly variable and related to a combination of histopathologic and genomic factors, phylogenetic analysis allows for the nomination of the likely clonal origin of LN disease. Despite the overall molecular diversity seen

within primary samples, there was no single identifiable factor in our analysis that was seen to be consistently associated with LN metastasis. Nevertheless, we found that a metastatic clone is frequently associated with EPE and/or cribriform pattern, suggesting that there are likely non-genomic, spatial, and/or anatomic determinants that contribute to LN metastasis. We also noted that the highest-grade region of the primary tumor does not always represent the metastatic clone, indicating that while histologic grade is generally associated with clinical outcomes, it is not necessarily the sole determinant of metastatic potential. Lastly, we observed significant heterogeneity in derived RNA-based prognostic scores within each profiled tumor region in individual patients as well as across the entire cohort. This suggests that prediction of which areas of a tumor will metastasize a priori using tools such as tissue-based prognostic biomarkers will remain challenging due to tumor heterogeneity. Taken together, our findings suggest prostate cancer LN metastasis is a complex event, with genomic, anatomic, and histopathological determinants all likely playing a role.

Our results are consistent with prior studies demonstrating the genomic heterogeneity of clinically localized prostate cancer[5,6,9]. These studies observed minimal shared CNAs and SNVs between disease foci, similar to our results which display a highly variable profile of CNAs among primary tumor samples, including samples obtained from different regions of a primary tumor as well as those obtained from multifocal disease. The relative molecular homogeneity of LN metastatic samples may represent an early step in monoclonal metastasis-to-metastasis seeding as has been previously demonstrated, or the emergence of seeding from a primary clone with metastatic potential[19]. Despite the molecular complexity of primary disease, most metastases in prostate cancer arise from a single precursor cell[20]. It is likely that this homogeneity then gives rise to clonal diversification as mutations accumulate, driving eventual clones towards a path of metastatic spread and treatment resistance[19]. Our work supports the presence of LN metastasis as a potential initial step for the development of distant metastases along this existing paradigm. Further, previous whole-genome sequencing efforts to characterize the lethal clone in a man with prostate cancer-related death have shown that the lethal clone may arise from a low-grade region within a high-grade primary cancer foci[12]. In our cohort, the biological clone likely giving rise to LN metastasis similarly did not always originate from the area of highest-grade disease. In at least three cases (patient #4, patient #34, patient #38) the primary tumor region that most resembled the LN metastatic foci was from a relatively lower-grade region of the primary tumor. In a previous paired analysis of 12 radical prostatectomy specimens from men who subsequently developed metastatic, castrate-resistant prostate cancer, truncal events included *SPOP* mutations, *TMPRSS2:ETS* gene rearrangements, and *TP53* mutations, which were all demonstrated to varying degrees in both primary and LN samples in our cohort[25]. Further, *PTEN*, *RB1*, *FANCA*, and/or *ATM* losses were similarly detected within primary and LN samples in our study, suggesting the presence of these alterations may confer increased aggressiveness and a propensity for the development of castration resistance.

There are several important clinical and research implications of the current study. First, the dilemma of reliably identifying the biologically dominant nodule presents a significant clinical challenge for risk stratification and treatment of primary prostate cancer. For example, as previously demonstrated by our group and others, tissue-based prognostic biomarkers may be of limited utility in determining the presence of unsampled aggressive disease[4,7]. Critically, the basic tenet of focal therapy for prostate cancer relies on the capacity to identify the cancer focus with the greatest metastatic potential for targeted ablation. Without consistent identification of the lesion most likely to be biologically harmful, there remains the possibility that such approaches subject men to treatment without oncological benefit if the focus with the most aggressive potential is missed. Second, our data demonstrate the inherent limitation of using molecular profiling

to determine the need for pelvic lymphadenectomy at the time of surgery or whole pelvis radiation with radiation therapy. As molecular profiling techniques continue to evolve and molecular data is increasingly available, it will be important to investigate how to better integrate clinicopathologic and genomic information into routine clinical practice paradigms. In our cohort, we observed LN metastasis in organ-confined disease in a patient with significant genomic complexity, with notable alterations in driver genes such as *FOXA1*, *ATM*, *RB1*, and *MYC*. Our work thus presents an initial step for the potential application of targeted molecular profiling to risk stratify patients with apparent organ-confined disease. An improved understanding of the molecular mechanisms underpinning LN metastasis in prostate cancer may facilitate the targeting of specific alterations or allow for tailored biomarkers that may lead to improvements in precision medicine.

The current study is not without limitations. First, we utilized targeted DNA and RNA sequencing using previously developed panels of genes with established relevance in prostate and other cancers. Notably, we demonstrated consistent results between targeted DNA NGS and LPWGS in a subset of samples, supporting the validity of our approach for CNA assessment. However, whole-genome or exome and transcriptome sequencing with greater depth is needed to capture potential targets that were unidentifiable with our targeted or LPWGS approaches. Further, evolving single-cell or spatial sequencing approaches may potentially provide greater insight into the existing heterogeneity of primary prostate cancer. Nevertheless, we were able to generate sufficient data to construct phylogenetic trees to determine the clonal evolution of primary tumor regions and nominate the area most likely to have given rise to LN metastases. Our approach was able to confirm the molecular heterogeneity present in primary prostate cancer and identify the relative molecular homogeneity of LN metastatic foci. Second, information regarding the laterality of LN metastasis in this cohort was not available, hence, conclusions regarding the propensity for lesions to spread in a unidirectional or bidirectional manner could not be ascertained. Lastly, it has been previously suggested that LN metastases are not necessarily an intermediate step for the development of distant metastasis, but a genetic dead-end in tumor phylogenies. Thus, future comparisons of paired LN metastasis with distant metastatic sites are needed, although access to such samples is limited. Despite these limitations, our comprehensive, multiregional, molecular profiling of primary prostate cancer and paired synchronous LN metastases sheds light on important histopathological factors and genomic alterations implicated in LN-positive disease.

The metastatic potential of prostate cancer results from a constellation of genomic alterations, molecular changes, and histopathological features, including tumor morphology, architecture, and anatomic location. Although phenotypic or histopathologic features are driven by their underlying genomic profile, the promise of molecular medicine lies in the ability to provide prognostic and predictive information beyond existing clinicopathologic data. Additional studies with emerging, more sophisticated molecular tools in larger cohorts are needed to more precisely identify primary foci most likely to cause harm and characterize the molecular underpinnings of LN metastasis.

## Methods

All study aspects were performed in accordance with the Good Clinical Practice Guidelines and Declaration of Helsinki and with local Institutional Review Board approval from the University of Michigan (IRB #: HUM00042749). Written informed consent was waived by the IRB for this retrospective analysis of archival tissue specimens.

### Patients and specimens

We assembled a retrospective, multi-institutional, non-consecutive cohort of men with primary prostate cancer with LN metastases. First, from a prior transcriptomic analysis of samples chosen to represent the clinical continuum of prostate cancer by our group[7], we identified *six* patients with synchronous LN metastases at the time of radical prostatectomy (*cohort 1*). Second, we assembled an additional non-consecutive cohort of *twelve* patients with large or multifocal prostate cancer with synchronous LN metastases from participating institutions (*cohort 2*). From a combined cohort of 18 patients (103 primary tumor and 28 LN metastasis samples), 10 patients (65 primary tumor and 16 LN metastasis samples) met experimental and analytic QC metrics as well as had sufficient quality data for phylogenetic analysis. Complete sample information is provided in supplementary file – **Source Data**. All profiled samples were evaluated by a board-certified Anatomic Pathologist (S.A.T., A.M.U., R.L.) with combined experience in prostate and molecular pathology who assigned a Gleason score, GG and outlined tumor regions within the specimens for molecular profiling. Prostate cancer is a disease of male sex, and all patients in this study self-reported as male.

### Hematoxylin and Eosin (H&E), Immunohistochemistry (IHC), DNA/RNA isolation

H&E–stained sections and ERG IHC images (as needed) were reviewed by board-certified anatomic pathologists (A.M.U., R.L., S.A.T.) to outline regions for multiregion molecular profiling. Punch biopsies (4–6 samples) of each region were obtained from formalin-fixed, paraffin-embedded (FFPE) radical prostatectomy blocks. For LN specimens, *ten* 10-µm serial sections of LN FFPE specimens were obtained from which macrodissection of outlined regions was performed with microscope guidance at 4x or 10x as needed. DNA and RNA were then co-isolated from these samples using the Qiagen Allprep FFPE DNA/RNA Kit (Qiagen, Hilden, Germany) and the Qiagen QIAcube (Qiagen, Hilden, Germany), according to the manufacturer's instructions. Extracted DNA and RNA samples were quantified using the Qubit 2.0 fluorometer (Life Technologies, California, United States).

### Targeted DNA NGS

Ion Torrent-based targeted DNA NGS was performed as described[26–28]. Briefly, NGS libraries were generated from up to 40 ng of FFPE-extracted DNA using the Ion AmpliSeq™ Library Kit 2.0 or Ion AmpliSeq™ Library Kit Plus (Thermo Fisher Scientific, Waltham, MA) and two AmpliSeq™ panels: the commercially available 409-gene Comprehensive Cancer Panel (CCP) and a custom 135-gene panel ("Pan-GU"). Barcoded NGS libraries were templated using an Ion OneTouch ES or Ion Chef Instrument (Thermo Fisher Scientific) prior to sequencing on an Ion Torrent Proton or S5 NGS System (Thermo Fisher Scientific). NGS reads were aligned to the human genome (hg19) using Ion Torrent Suite (Thermo Fisher Scientific); NGS libraries were subjected to standard post-sequencing QC measures (average depth >350X and uniformity >70%); and variants were identified using the variantCaller plugin. Identified variants were annotated using ANNOVAR[29] and filtered using standard criteria to remove FFPE and sequencing artifacts: flow-corrected variant allele-containing reads (FAO) less than 6; variant allele frequencies (VAF; FAO/FDP) less than 0.05; and, skewed variant read support [>five-fold difference in the number of forward (FSAF) versus reverse (FSAR) reads containing the variant allele (FSAF/FSAR < 0.2 or FSAF/FSAR > 5)]. Germline variants were removed by filtering tumor samples against matched normal samples, and synonymous, intronic, and intergenic variants were excluded from subsequent analyses. Finally, all passed variants were manually visualized and confirmed using the Integrative Genomics Viewer (Broad Institute, Cambridge, MA) by an experienced molecular pathologist (A.M.U.).

### Copy number alterations (CNA) analyses

CNA analysis was performed on targeted DNA NGS data sequenced on Ion Torrent Proton or S5 NGS systems[23]. Briefly, for each tumor sample, raw amplicon-level NGS read counts were normalized, GC content corrected and then compared to matched normal samples to

determine a copy number ratio for each amplicon. We used hg38 as the reference genome to map tumor and normal reads. Gene-level CNA estimates were derived by taking the coverage-weighted mean of the amplicon ratios. $Log_2$ copy number ratios were then generated for each gene and the variability of amplicons within each gene was used to determine q-values. Gene level losses and gains were determined based on the derived q-values. We used a custom quality control filtering criteria which retained the top 445 genes that were retained on the displayed heatmap.

### Targeted RNA NGS and scores derivation

Ion Torrent-based targeted RNA NGS libraries[7]. Briefly, NGS libraries were generated from up to 20 ng of FFPE-extracted RNA using the Ion AmpliSeq™ Library Kit 2.0 or Ion AmpliSeq™ Library Kit Plus (Thermo Fisher Scientific) and a custom prostate cancer-focused AmpliSeq™ panel comprising 306 amplicons targeting known gene fusions and prostate cancer relevant transcripts/pathways. Barcoded NGS libraries were templated using an Ion OneTouch ES or Ion Chef Instrument (Thermo Fisher Scientific) prior to sequencing on an Ion Torrent Proton or S5 NGS System (Thermo Fisher Scientific). NGS reads were aligned to the targeted transcripts using Ion Torrent Suite (Thermo Fisher Scientific). Four of 5 Oncotype DX panel housekeeping genes (ATP5E, ARF1, CLTC1, and PGK1) with robust expression were used for normalization prior to downstream analyses. Data processing, sample filtering, normalization, and QC steps were performed in the *R* project for Statistical Computing v. 3.4.3. Samples with at least 500,000 total mapped sequencing reads and at least 60% of all end-to-end reads were retained. We have previously described our sample exclusion criteria based on housekeeping gene expression and calculation of prognostic scores (mxCCP, mxGC, and mxGPS)[7]. A sample was classified as gene fusion-positive based on RNA NGS data if the fusion partner count was ≥500 and $log_2$ normalized partner expression was ≥−6 ($log_2(0-01)$). For one sample without sufficient RNA to determine gene fusion status, we used ERG expression on IHC to classify it into fusion-positive vs. negative. Notably, we have previously reported 100% concordance between RNA NGS- and ERG IHC-based approaches for determining gene fusion status[30].

### Low-pass whole-genome sequencing (LPWGS)

LPWGS libraries were constructed from 100 ng of FFPE-extracted DNA using the Twist Enzymatic Fragmentation 2.0 Library Preparation Kit (Twist Biosciences, South San Francisco, CA) and sequenced in 300-cycle paired-end read format using an Illumina NextSeq 2000 (Illumina, San Diego, CA). Raw sequencing data was processed using Illumina DRAGEN software and aligned to the human genome (hg38) using standard settings for tumor-normal pairs with the DRAGEN DNA Pipeline. The median total sequenced reads per library was 31,988,813 (range = 17,734,684−38,444,792), corresponding to a median maximum genomic coverage of 1.12X (range = 0.64−1.37X). Output bigWig files were then utilized as input for CNA using ichorCNA with standard settings[31]; normal contamination was estimated for each sample using prioritized variant information from targeted DNAseq data (Patient #1, P1-P4 and LN1-2) or set to default (0.5; P5). For comparison to targeted DNAseq data, segmented copy number data was visualized as log2 copy number ratios with a minimum threshold of +/− 0.3 (Fig. S2).

### Phylogenetic analyses

To reconstruct the evolutionary trajectory of the primary tumor samples in relation to synchronous LN metastases, we utilized two methods, namely, neighbor joining and PhyloWGS methods.

### Neighbor-joining method

For each tumor sample within a patient, somatic alterations, significant copy number alterations and gene fusion status, as determined above, were considered as features for evolutionary analysis to determine clonality. We constructed distance matrices using the neighbor-joining algorithm by feeding the features as present or absent using first order model of variances and covariances of the evolutionary distance estimates. At each step, the model selects from a class of admissible reductions which minimizes the variance of the new distance matrix. Unrooted trees derived from the resulting matrices were converted into phylogenetic trees using *R* package *phangorn*. Primary tumors branched with LN tumors were considered to have evolved from same parent clone during progression. This approach was utilized for all patients included in the cohort.

### PhyloWGS method

To create simple somatic mutation (SSM) and copy number variation (CNV) files compatible for PhyloWGS, an in-house script was developed for use on the Ion Torrent NGS data. Briefly, circular binary segmentation from the R package DNA NGS was run on GC-corrected and normalized amplicon-level log2 ratios (log2R), and a z-score was calculated to compare tumor to normal samples. Genomic segments were filtered by size and q-value prior to conversion from log2R to integer values. For this conversion, a modified allele specific copy number analysis of tumors (ASCAT) equation was utilized: $(na + nb) = (2^{(log2R)} * pl − 2 * (1-p)) / p$, with pl being ploidy, p being purity, na and nb being maternal and paternal copy number, respectively. All tumors were assumed to be diploid, and tumor purity was estimated by an anatomic pathologist (with a floor of 0.5 to avoid over-scaling in the equation). Afterwards, variants from the SSM file were intersected with the segments, and paternal and maternal copy-numbers were assigned to the CNV file. These assignments were based on each sample's presence or absence of an overlapping CNV and variant fraction from the SSM file. PhyloWGS was then run with default parameters using the multievolve Python script, and the tree selected was based on PhyloWGS's metric of "best tree". Multiple primary trees were allowed in cases where the algorithm flagged for it. Cellular prevalence (**Source Data**) was then used to visualize where the samples split based on the subclone phylogeny tree. This approach was utilized for patients #1, 33, and 41 as shown in Fig. S3 to demonstrate concordance with the neighbor joining method as shown in Fig. 2.

### Reporting summary

Further information on research design is available in the Nature Portfolio Reporting Summary linked to this article.

## Data availability

The bam files with raw targeted RNA NGS and LPWGS data generated in this study have been deposited to the European Nucleotide Archive (ENA) database under accession code PRJEB72307. The data is publicly available, and access can be obtained at ENA website. Source data are provided with this paper.

## Code availability

The *R* code used for the analysis has been deposited to github [https://github.com/srinew/PCA-LNMETS].

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

## Acknowledgements

U.S. is supported by a Urology Care Foundation – American Urological Association Research Scholar Award sponsored by the Society of Urologic Oncology/Specialized Programs of Research Excellence (SPORE), a Prostate Cancer Foundation Young Investigator Award, a NIH Loan Repayment Program Award (L30 CA264387), and SPORE Career Enhancement Program Award. U.S. and J.J.T. are supported by the Scholars Awards grants program of Precision Health at the University of Michigan. This work is also supported in part by the Prostate Cancer Foundation (U.S., J.T.T., S.A.T., S.S.S., and T.M.M.) and the National Institutes of Health (U01 CA214170 to S.A.T., R01 CA183857 to S.A.T., R37CA283857 to S.S.S., the University of Michigan Prostate S.P.O.R.E., P50 CA186786-05: U.S., A.M.U., G.S.P., S.S.S., and T.M.M). J.J.T. has been supported by the SPORE Career Enhancement Program (CA186786), the Prostate Cancer Foundation Young Investigator Award (20YOUN11), and an NIH Early Detection Research Network Biomarker Characterization Center award (U2CCA271854). Supported by the Men of Michigan Prostate Cancer Research Fund and the University of Michigan Comprehensive Cancer Center core grant (2-P30-CA-046592-24; S.S.S. and A.M.U.). T.M.M. and S.A.T. have been supported by the A. Alfred Taubman Biomedical Research Institute. S.S.S. is supported by the Department of Defense (W81XWH1810219), Robert Wood Johnson Foundation as part of the Harold Amos Medical Faculty Development Program (AMFDP), the Urology Care Foundation Rising Stars in Urology Research Award Program and Astellas, Inc. A.M.U. is supported by the Department of Defense (W81XWH-19-1-0407 and W81XWH-21-1-0238).

## Author contributions

All authors contributed extensively to the work presented in this paper. U.S., G.S.P., A.M.U., and S.S.S. jointly conceptualized the study. Analysis was performed by S.N. and K.H., with input and assistance from U.S., J.J.T., T.M., M.C., A.M.C., S.A.T., G.S.P., A.M.U., and S.S.S. Review of pathology was conducted by S.A.T, R.L. and A.M.U. Samples were provided by J.S.K., T.M., R.L., S.F.S., N.F., L.D., C.D., A.B., and G.S.P. T.M.P., J.S.K., C.J.L., R.K., K.R.P., assisted with specimen processing, RNA/DNA isolation, and sequencing experiments. The manuscript was written by U.S., S.N., A.M.U., G.S.P., and S.S.S, with input from all authors. All authors agreed on the content and submission of the manuscript.

## Competing interests

The content is solely the responsibility of the authors and does not necessarily represent the official views of U-M Precision Health. JJT is a co-founder with minor equity interest in LynxDx, which has licensed urinary biomarkers unrelated to this project. S.A.T has received travel support from and had a sponsored research agreement with Compendia Bioscience/Life Technologies/ThermoFisher. The University of Michigan and Brigham and Women's Hospital have been issued a patent on *ETS* gene fusions (US8969527B2 "Recurrent gene fusions in prostate cancer") in prostate cancer on which S.A.T. and A.M.C. are co-inventors. The diagnostic field of use was licensed to Hologic/Gen-Probe, Inc., which

has sublicensed rights to Roche/Ventana Medical Systems. S.A.T. has served as a consultant for and received honoraria from Janssen, AbbVie, Sanofi, Almac Diagnostics and Astellas/Medivation. S.A.T. has sponsored research agreements with Astellas and GenomeDX. S.A.T. is a co-founder and Chief Medical Officer for Strata Oncology. T.M.M. has received research funding from MDxHealth, Myriad Genetics, and GenomeDx. T.M.M. has served as a consultant for Myriad Genetics. U.S. has received research funding from Merck & Co., Inc. S.S.S. is on a study advisory committee for Bayer Pharma and has a non-sponsored research agreement with GenomeDx. The other authors have declared that no conflict of interest exists.
