## [Peer Review File · Nature Communications]

REVIEWER COMMENTS

Reviewer #2 (Remarks to the Author): Expert in cancer genomics and evolution, and prostate cancer

The study by Singhal et al used multi-region prostate cancer samples from ten patients with lymph node metastases to perform targeted NGS, which they used to construct simple phylogenetic trees. The study aims to identify the primary lesions most likely to metastasize.

The study confirms previously reported heterogeneity of primary tumours and shows that lymph node metastatic samples display molecular homogeneity but without any common features or driver mutations. The authors confirmed that a metastatic clone does not necessarily resemble the most aggressive area, according to the Gleason score. The authors also note that EPE and cribriform patterns were often observed in areas of likely dissemination, which is interesting, but will need a larger dataset to confirm.

Studying primary sites leading to metastasis as well as primary and synchronous lymph nodes is relevant. However, there are many significant shortcomings of the study, the findings appear mostly incremental and there is a lack of novelty. The method used to reconstruct phylogenies rely on oversimplifications (see below). In combination with the likely poor quality of FFPE material and relatively few mutations makes analysis difficult to ascertain.

Main comments

- Insufficient/poorly detailed methods section: Three different panels are used to perform targeted sequencing. The methods section is very limited and no details are provided on the QC of the variants called, overlap and sensitivity of the different panels (each will have their own biases) and the variant filtering. Variant calling and QC is poorly described, e.g. "Annotated variants were filtered to remove poorly supported calls/sequencing..." ^[1]QC is particularly important when using FFPE material, which is more prone to false negatives (missing signal) and false positives (artefact mutations induced by formalin fixation).
- CNA calling from panel sequencing is highly challenging and needs to be compared/benchmarked with samples containing both panel and WGS. Finding >50% CDKN2B copy number changes is surprising and not consistent with the literature.
- The phylogenetic analysis is too simplistic and prone to misinterpretations. VAF is not used, despite the expected high coverage (I could not find information on sequencing coverage). Mutation calls are binarised, thereby removing crucial information on clonality (VAF/CCF) from the individual SNVs.

Identification of serial seeding, e.g. patient #2, #30, can be wrongly assigned if overcalling mutations, and need careful validation.

- The panels are focused on known cancer driver mutations and finding that driver mutations are shared between primary and relapse is not novel, nor surprising (summary of the third chapter, line 243).

minor

- The abstract mentions 14 patients, but in reality, only 10 patients were used for analysis.

- a typo has converted numbers to date in line 129.

Reviewer #3 (Remarks to the Author): Clinical expert in prostate cancer, urology, and lymph node metastases

First, I would like to congratulate the authors on their excellent elaborate work and subsequently drafted well-written descriptive manuscript. It delivers an insight in the background of LN metastases in prostate cancer that potentially impacts the clinical field with the emergence of treatment strategies such as focal ablative therapies and tailored triage to pelvic lymph node dissection. However I do miss one important, possibly very meaningful, part of the clinical work-up. I would greatly recommend to incorporate radiomics (such as derived from mpMRI and PSMA-PET scanning) in the analysis as clinical variables. However I would understand that these data are not available for (all) these patients, but then would at least suggest to include them as possible important clinical variables in addition to the histological stratifiers in the discussion.

Reviewer #2 (Remarks to the Author)

The study by Singhal et al used multi-region prostate cancer samples from ten patients with lymph node metastases to perform targeted NGS, which they used to construct simple phylogenetic trees. The study aims to identify the primary lesions most likely to metastasize. The study confirms previously reported heterogeneity of primary tumours and shows that lymph node metastatic samples display molecular homogeneity but without any common features or driver mutations. The authors confirmed that a metastatic clone does not necessarily resemble the most aggressive area, according to the Gleason score. The authors also note that EPE and cribriform patterns were often observed in areas of likely dissemination, which is interesting, but will need a larger dataset to confirm. Studying primary sites leading to metastasis as well as primary and synchronous lymph nodes is relevant. However, there are many significant shortcomings of the study, the findings appear mostly incremental and there is a lack of novelty. The method used to reconstruct phylogenies rely on oversimplifications (see below). In combination with the likely poor quality of FFPE material and relatively few mutations makes analysis difficult to ascertain.

Main comments

- Insufficient/poorly detailed methods section: Three different panels are used to perform targeted sequencing. The methods section is very limited and no details are provided on the QC of the variants called, overlap and sensitivity of the different panels (each will have their own biases) and the variant filtering. Variant calling and QC is poorly described, e.g. "Annotated variants were filtered to remove poorly supported calls/sequencing...".. QC is particularly important when using FFPE material, which is more prone to false negatives (missing signal) and false positives (artefact mutations induced by formalin fixation).

Authors' Response: We thank the reviewer for the thorough review of our manuscript and the comments provided. Together with Dr. Scott Tomlins (prior to his departure for industry), our group pioneered the application of Ion Torrent-based targeted next generation sequencing (NGS) to Formalin-Fixed Paraffin-Embedded (FFPE) material, and we have extensive experience with variant identification, filtering, and curation – having analyzed over 3,000 FFPE samples using this approach. Nonetheless, we acknowledge the issues raised by the reviewer and have updated the methods to provide detailed information regarding NGS variant calling. The revised text for "**Targeted, multiplex DNA NGS**" in the methods section is as follows:

Targeted DNA NGS: Ion Torrent-based targeted DNA NGS was performed as described (PMID: 25925381, 28403382, and 31969336). Briefly, NGS libraries were generated from up to 40 ng of FFPE-extracted DNA using the Ion AmpliSeq™ Library Kit 2.0 or Ion AmpliSeq™ Library Kit Plus (Thermo Fisher Scientific, Waltham, MA) and two AmpliSeq™ panels: the commercially available 409-gene Comprehensive Cancer Panel (CCP) and a custom 135-gene panel ("Pan-GU"). Barcoded NGS libraries were templated using an Ion OneTouch ES or Ion Chef Instrument (Thermo Fisher Scientific) prior to sequencing on an Ion Torrent Proton or S5 NGS System (Thermo Fisher

Scientific). NGS reads were aligned to the human genome (hg19) using Ion Torrent Suite (Thermo Fisher Scientific); NGS libraries were subjected to standard post-sequencing QC measures (average depth >350X and uniformity >70%); and variants were identified using the variantCaller plugin. Identified variants were annotated using ANNOVAR (PMID: 20601685) and filtered using standard criteria to remove FFPE and sequencing artifacts: flow-corrected variant allele-containing reads (FAO) less than 6; variant allele frequencies (VAF; FAO/FDP) less than 0.05; and, skewed variant read support [$>$ five-fold difference in the number of forward (FSAF) versus reverse (FSAR) reads containing the variant allele ($FSAF/FSAR < 0.2$ or $FSAF/FSAR > 5$)]. Germline variants were removed by filtering tumor samples against matched normal samples, and synonymous, intronic, and intergenic variants were excluded from subsequent analyses. Finally, all passed variants were manually visualized and confirmed using the Integrative Genomics Viewer (Broad Institute, Cambridge, MA) by an experienced molecular pathologist (A.M.U.).”

Also, please note that the methods section from the original manuscript submission inadvertently stated that a third DNA NGS panel (OCP) was utilized. A few of the samples were initially sequenced using the OCP panel prior to the decision to sequence all samples with both the CCP and Pan-GU panels; however, all data presented in the manuscript are only derived from the CCP and Pan-GU panels, which provide a good balance between comprehensive genomic sequencing (CCP) and prostate cancer-specific genomic alterations (Pan-GU). This has been clarified in the above revised text.

- CNA calling from panel sequencing is highly challenging and needs to be compared/benchmarked with samples containing both panel and WGS. Finding >50% CDKN2B copy number changes is surprising and not consistent with the literature.

Authors' Response: We thank the reviewer for this comment and acknowledge that CNA analysis from targeted NGS data can be challenging; however, as described above, our group has extensive experience with these methods and have previously benchmarked CNA data from targeted NGS to fluorescence in situ hybridization (FISH), comparative genomic hybridization (CGH) array, and WES data, showing a high degree of concordance among these approaches (PMID: 25468433). Nonetheless, to further address these concerns, we performed low-pass whole-genome sequencing (LPWGS) on samples from Patient-#1 (n = 8 samples, including the normal sample) from our cohort. Notably, we observed concordance between CNA from our targeted NGS approach to CNA from LPWGS (**Fig. S2**). Additionally, in an unsupervised hierarchical clustering analysis, samples from primary prostate cancer regions P1 and P2 clustered together with lymph node metastatic regions LN1 and LN2 as observed in our phylogenetic tree analyses using our targeted sequencing approach.

Regarding the observation of >50% CDKN2B copy number changes in our cohort, it is important to recognize that these are single-copy loss events in the context of chromosome 9p21 aneuploidy – not deep deletions (i.e., two-copy loss events). Furthermore, the provided percentage is misleading because it was calculated from the total number of samples and doesn't account for multiple clonally related samples from

a single patient. Therefore, to prevent confusion, we have removed these percentages from the manuscript and figures.

- The phylogenetic analysis is too simplistic and prone to misinterpretations. VAF is not used, despite the expected high coverage (I could not find information on sequencing coverage). Mutation calls are binarised, thereby removing crucial information on clonality (VAF/CCF) from the individual SNVs. Identification of serial seeding, e.g. patient #2, #30, can be wrongly assigned if overcalling mutations, and need careful validation.

Authors' Response: We thank the reviewer for this comment and acknowledge that phylogenetic reconstruction from targeted NGS data can be challenging – particularly for tumors with relatively low mutational burden (such as primary prostate cancer). Our group previously utilized a parsimonious approach with the dollop method in the Phylogeny Inference Package (PHYLIP) to determine multiclonality and clonal evolution of endometrial carcinomas using variant data from targeted NGS of FFPE tissue (PMID: 30610106). Given the relatively low mutation burden of primary prostate cancer in general and as observed in this study, we sought to expand the number of genomic events to be considered in the phylogenetic reconstruction by also incorporating CNA data parsimoniously in a similar manner using the neighbor-joining algorithm in the phangorn package. Since this approach does not utilize copy number-normalized VAF to estimate the cancer cell fraction (CCF) for a given variant, we acknowledge that some potential information may not be fully accounted for in the clonality assessment. That being said, manual review of the constructed phylogenetic trees – considering the underlying identified variants and CNA – suggests that the phylogenetic reconstructions make sense intuitively.

To further address these concerns, we have utilized the PhyloWGS approach, a method that accounts for variant fractions to generate phylogenetic reconstructions for several of our samples and overall observed similar results as shown in **Fig. S3**. One limitation of the PhyloWGS approach is that it does not account for RNA level data input, thus we were unable to utilize TMPRSS2:ERG gene fusion status, a known early clonal event in prostate cancer, in the phylogenetic tree reconstruction analyses. Notwithstanding, we observed high concordance between the phylogenetic trees constructed using our parsimonious approach and those constructed using the PhyloWGS method. This data is presented for patients #1, #33, #41 (**Fig. S3**) as a supplement to **Fig. 2**. The corresponding cellular prevalence and cancer cell fraction (CCF) making up each node by sample are shown in **Table S2**.

- The panels are focused on known cancer driver mutations and finding that driver mutations are shared between primary and relapse is not novel, nor surprising (summary of the third chapter, line 243).

Authors' Response: While we agree with the reviewer that the finding that driver mutations are shared between primary and metastatic sites is not novel, we would contend that the idea of identifying the location of the primary lesion that gives rise to synchronous lymph node metastasis is novel, which is the focus of the current study.

These findings, if validated across larger datasets with high-depth sequencing, would have important clinical implications. For example, if a molecular analysis of primary, multifocal prostate cancer would allow for identification of the lesion most likely to give rise to metastasis, this would allow for potential utility of focal therapy interventions (cryoablation, radiofrequency ablation, HIFU) targeted to this specific lesion. Therefore, we believe this work would serve as the basis for future studies that may be able to further characterize the importance of understanding the molecular basis of primary, multifocal prostate cancer with synchronous nodal metastases.

minor

- The abstract mentions 14 patients, but in reality, only 10 patients were used for analysis.
- a typo has converted numbers to date in line 129.

Authors' Response: The above minor comments have been edited and addressed in the text of the manuscript. Thanks for pointing these out.

Reviewer #3 (Remarks to the Author): Clinical expert in prostate cancer, urology, and lymph node metastases

First, I would like to congratulate the authors on their excellent elaborate work and subsequently drafted well-written descriptive manuscript. It delivers an insight in the background of LN metastases in prostate cancer that potentially impacts the clinical field with the emergence of treatment strategies such as focal ablative therapies and tailored triage to pelvic lymph node dissection. However, I do miss one important, possibly very meaningful, part of the clinical work-up. I would greatly recommend incorporating radiomics (such as derived from mpMRI and PSMA-PET scanning) in the analysis as clinical variables. However, I would understand that these data are not available for (all) these patients, but then would at least suggest to include them as possible important clinical variables in addition to the histological stratifiers in the discussion.

Authors' Response: We would like to thank this reviewer for their comments regarding our manuscript. We agree that incorporating radiomics would be extremely beneficial to allow for correlation of molecular findings with radiographic characteristics in primary, multifocal prostate cancer. Unfortunately, clinical data with regards to imaging was not available for the patients in the current study. Certainly, future analyses incorporating imaging characteristics with molecular findings to identify molecular characteristics and drivers of synchronous lymph node metastasis in multifocal disease will be informative.

REVIEWERS' COMMENTS

Reviewer #2 (Remarks to the Author):

The authors have responded to the main quality-based issues raised and the lack of details in the Methods description.

The manuscript is, however, still mostly descriptive, is based on low-confidence variant calls (panel seq from FFPE) and lacks novelty.

Reviewer #3 (Remarks to the Author):

My concerns were adequately addressed.

REVIEWERS' COMMENTS

Reviewer #2 (Remarks to the Author):

The authors have responded to the main quality-based issues raised and the lack of details in the Methods description.

The manuscript is, however, still mostly descriptive, is based on low-confidence variant calls (panel seq from FFPE) and lacks novelty.

We thank this reviewer for their review of our manuscript. We hope our results add to the growing literature regarding clonality in prostate cancer, and look forward to additional, large-scale studies corroborating our findings.

Reviewer #3 (Remarks to the Author):

My concerns were adequately addressed.

We thank this reviewer for their review of our manuscript.